# FILTRA: Rethinking Steerable CNN by Filter Transform

## Abstract

Steerable CNN imposes the prior knowledge of transformation invariance or equivariance in the network architecture to enhance the the network robustness on geometry transformation of data and reduce overfitting. Filter transform has been an intuitive and widely used technique to construct steerable CNN in the past decades. Recently, group representation theory is used to analyze steerable CNN and reveals the function space structure of a steerable kernel function. However, it is not yet clear on how this theory is related to the filter transform technique. In this paper, we show that kernel constructed by filter transform can also be interpreted in the group representation theory. Meanwhile, we show that filter transformed kernels can be used to convolve input/output features in different group representation. This interpretation help complete the puzzle of steerable CNN theory and provides a novel and simple approach to implement steerable convolution operators. Experiments are executed on multiple datasets to verify the feasibilty of the proposed approach.

## 1 Introduction

Beyond the well-known property of equivariance under translation, there has been substantial recent interest in CNN architectures that are equivariant with respect to other transformation groups, e.g. reflection and rotation. Applications of such architectures range over scenarios where object orientation might variate, including OCR, aerial image processing, 3D point cloud processing, medical image processing, texture analysis and etc.

Previous works on constructing equivariant CNN can be coarsely categorized as two aspects. The first aspect designs special *steerable filters* so that the convolutional output is hard-baked to transform accordingly when the input reflects or rotates. A plenty of works develop this idea by filter rotation, including hand-crafted filters (Oyallon & Mallat, 2015) and learned filters (Laptev et al., 2016; Zhou et al., 2017; Cheng et al., 2018; Marcos et al., 2017). TI-Pooling (Laptev et al., 2016) produce invariant output as input rotates. ORN (Zhou et al., 2017) and RotDCF (Cheng et al., 2018) produces output which circularly shifted as input rotates. Since each dimension of such permutable output corresponds to a uniformly discrete rotation angle, RotEqNet (Marcos et al., 2017) propose to extract rotation angle from the permutable features. Another approach to construct steerable filters is to linearly combine a set of steerable bases. These basis can be solved in discrete function space (Cohen & Welling, 2014; 2016) or continuous function space (Worrall et al., 2017; Weiler & Cesa, 2019). Weiler & Cesa (2019) comprehensively summarize works on steerable bases using polar Fourier basis.

The second aspect exploits specific transforms to act on input. Spatial Transformer Network (STN) is a well-known representative, which predicts an affine matrix to transform its input to the canonical form. Tai et al. (2019) inherits this idea to design equivariant network. Another choice of transform is to the polar coordinate system (Henriques & Vedaldi, 2017; Esteves et al., 2018). Since 2D rotation in Cartesian coordinate system corresponds to 2D translation in polar coordinate system, rotation equivariance can be achieved by conventional translation equivariant CNN.

The approach proposed in this paper falls into the first category. Weiler & Cesa (2019) proves that all steerable convolutional operator could be denoted as the combination of a specific set of polar Fourier bases. However, it is not clear yet how this interpretation is related with the widely used filter transform scheme. In this paper, we aim to establish the missing connection between the

group representation based analysis for steerable filters and filter transform scheme. To this end, we propose a new approach (FILTRA) to use filter transform to establish steerability between features in different group representation in cyclic group $C_N$ and dihedral group $D_N$. We verify the feasibility of FILTRA for the classification and regression tasks on different datasets.

## 2 PRELIMINARIES

We make use of several NumPy or SciPy functions in equations including `roll`[1], `flipud`[2] and `circulant`[3]. We omit the variable in bracket sometimes by writing $\kappa_*^* = \kappa_*^*(g)$ and $K_*^* = K_*^*(\phi)$.

### 2.1 STEERABLE CNN

We recapitulate the basic concepts of steerable CNN which will be frequently used in this paper. For detailed introduction, readers can refer to Weiler & Cesa (2019) for a comprehensive information. We mainly consider the 2D image case and denote $x \in \mathbb{R}^2$ as a pixel coordinate. We use vector field $f(x) \in \mathbb{R}^C$ to denote a general multi-channel image, where $C$ is the number of channels. Typical examples of $f(x)$ include RGB image $f(x) \in \mathbb{R}^3$ and gradient image $f(x) \in \mathbb{R}^2$. Consider a group $G$ of transformations and an element $g \in G$. Examples of $G$ include rotation, translation and flip. A vector field $f(x)$ follows the below rules when undergoing the act $\pi(g)$ of a group element $g$:

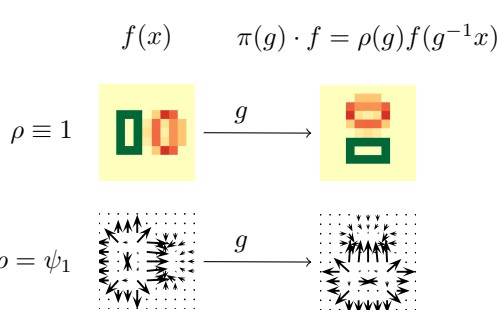

$$\pi(g) \cdot f = \rho(g)f(g^{-1}x), \tag{1}$$

where $\rho(g)$ is a group representation related to vector field $f$. Fig. 1 shows an example of different types of $\rho$ for RGB images and gradient images under a rotation transform element $g$. The group representation of RGB is $\rho(g) \equiv 1$ while for gradient image $\rho(g)$ is a 2D rotation matrix which also rotates vector $f(x)$ by $g$.

Figure 1: Examples of images (feature maps) with different group representation $\rho$. Both images undergo 90deg rotation. The upper row is an RGB image whose 3-channel colors remain the same when the image is rotated. The lower row is a gradient image whose two channel value should be rotated in the same way when the gradient image is rotated.

In the scenario of convolutional neural network, a convolution operator $f \mapsto \kappa \cdot f$ is considered as *steerable* if it satisfies

$$\kappa \cdot [\pi_1(g)f] = \pi_2(g)[\kappa \cdot f], \tag{2}$$

i.e. the output vector field transforms equivariantly under $g$ when the input is transformed by $g$.

### 2.2 REFLECTION GROUP, CYCLIC GROUP AND DIHEDRAL GROUP

We consider steerable filters on reflection group $(\{\pm 1\}, *)$, cyclic group $C_N$ and dihedral group $D_N = (\{\pm 1\}, *) \ltimes C_N$. To unify the notations in derivation, we interpret $C_N = (\{1\}, *) \ltimes C_N$ and $(\{\pm 1\}, *) = (\{\pm 1\}, *) \ltimes C_1 = D_1$ so that a element in these three groups can always be denoted as a pair $g = (i_0, i_1)$, whose range is $\mathbb{Z}_2 \times \mathbb{Z}_1$ for reflection group, $\mathbb{Z}_1 \times \mathbb{Z}_N$ for cyclic group and $\mathbb{Z}_2 \times \mathbb{Z}_N$ for dihedral group. Each element in $C_N$ corresponds to rotation angle $\theta_{i_1} = \frac{2i_1\pi}{N}$.

### 2.3 GROUP REPRESENTATION

A linear representation $\rho$ of a group $G$ on a vector space $\mathbb{R}^n$ is a group homomorphism from G to the general linear group $GL(n)$, denoted as

$$\rho : G \mapsto GL(n) \quad \text{s.t.} \quad \rho(gg') = \rho(g)\rho(g'), \quad \forall g, g' \in G. \tag{3}$$

---

[1]`https://numpy.org/doc/stable/reference/generated/numpy.roll.html`
[2]`https://numpy.org/doc/stable/reference/generated/numpy.flipud.html`
[3]`https://docs.scipy.org/doc/scipy/reference/generated/scipy.linalg.circulant.html`

We consider three types of linear representation in this paper, i.e. trivial representation, regular representation and irreducible representation (irrep). Readers can refer to Serre (1977) for further background for these concepts.

The trivial representation of a group element is always $\rho_{\text{tri}}(g) \equiv 1$. The regular representation of a finite group $G$ acts on a vector space $\mathbb{R}^{|G|}$ by permuting its axis. Therefore, for a rotation element $g = (0, i_1) \in C_N$ or $D_N$, we get

$$\rho_{\text{reg}}^{C_N}(g) = P(i_1), \quad \rho_{\text{reg}}^{D_N}(g) = \begin{bmatrix} P(i_1) & 0 \\ 0 & P(i_1) \end{bmatrix}, \text{where} \quad P(i_1) = \texttt{roll}(\mathbf{I}_N, i_1, 0). \quad (4)$$

For a reflected element $g = (1, i_1) \in D_N$, we get

$$\rho_{\text{reg}}^{D_N}(g) = \begin{bmatrix} 0 & B(i_1) \\ B(i_1) & 0 \end{bmatrix}, \text{where} \quad B(i_1) = \texttt{flipud}(P(-i_1 - 1)). \quad (5)$$

By selecting suitable change of basis of the vector space, a representation can be converted to a equivalent representation, which is the direct sum of several independent representations on the orthogonal subspaces. A representation is called irreducible representation if no non-trivial decomposition exists. This conversion is denoted as

$$\rho(g) = Q \left[ \bigoplus_{(i_0, i_1) \in I} \psi_i(g) \right] Q^{-1}, \quad (6)$$

where $I$ is an index set specifying the irreducible repsentations $\psi_i$ and $Q$ is the change of basis.

## 2.4 DECOMPOSING REGULAR REPRESENTATION

We decompose the regular representation into a set of irreps. Define the following base irrep

$$\psi_{j,k}(i_0, i_1) = \begin{cases} ((-1)^j)^{i_0} & k = 0 \\ (-1)^{i_1} \cdot ((-1)^j)^{i_0} & k = \frac{N}{2}, N \text{ is even} \\ \begin{bmatrix} \cos(k\theta_{i_1}) & -\sin(k\theta_{i_1}) \\ \sin(k\theta_{i_1}) & \cos(k\theta_{i_1}) \end{bmatrix} \begin{bmatrix} 1 & 0 \\ 0 & (-1)^{i_0} \end{bmatrix} \cdot ((-1)^j)^{i_0} & \text{otherwise} \end{cases}, \quad (7)$$

where $j, k$ are referred as the reflection and rotation frequency of the irrep. Concretely, if the action $g$ reflects/rotates an object once, $\psi_{j,k}(g)$ will reflects/rotates in vector space $j/k$ times. We also define the following discrete cosine transform basis

$$V = \begin{bmatrix} \beta_0 & \beta_1 & \cdots & \beta_{\lfloor \frac{N}{2} \rfloor} \end{bmatrix},$$

$$\text{where} \quad \beta_k = \begin{cases} \mathbf{1}_N & k = 0 \\ \begin{bmatrix} \cos(k\theta_0) & \cos(k\theta_1) & \cdots & \cos(k\theta_{N-1}) \end{bmatrix}^\top & k = \frac{N}{2}, N \text{ is even} \\ \begin{bmatrix} \cos(k\theta_0) & \cos(k\theta_1) & \cdots & \cos(k\theta_{N-1}) \\ \sin(k\theta_0) & \sin(k\theta_1) & \cdots & \sin(k\theta_{N-1}) \end{bmatrix}^\top & \text{otherwise} \end{cases}. \quad (8)$$

The following decomposition for $\rho_{\text{reg}}^{C_N}(0, i_1)$ holds

$$\rho_{\text{reg}}^{C_N}(g) = V D^{C_N} V^\top, \quad D^{C_N} = \bigoplus_{0 \le k \le \lfloor \frac{N}{2} \rfloor} \psi_{0,k}(0, i_1). \quad (9)$$

The decomposition for $\rho_{\text{reg}}^{D_N}(i_0, i_1)$ holds in a bit more complicated form, i.e.

$$\rho_{\text{reg}}^{D_N}(i_0, i_1) = W D^{D_N} W^\top,$$

$$\text{where} \quad W = \begin{bmatrix} V & V \\ V & -V \end{bmatrix}, \quad D^{D_N} = \bigoplus_{0 \le j \le 1, 0 \le k \le \lfloor \frac{N}{2} \rfloor} \psi_{j,k}(i_0, i_1), \quad (10)$$

and each column of $W$ is refered by $\beta_{j,k} = \begin{bmatrix} \beta_k^\top & (-1)^j \beta_k^\top \end{bmatrix}^\top$. See Fig. 2 for a visualization of this decomposition.

We also mention a property of $\beta_k$ that is easy to verify and will be useful in our derivation.

$$\psi_{0,k}(0, i_1)\beta_k^\top = \beta_k^\top P(i_1), \quad \psi_{1,k}(0, i_1)\beta_k^\top = \beta_k^\top P(i_1), \quad (11a)$$

$$\psi_{0,k}(1, i_1)\beta_k^\top = \beta_k^\top B(i_1), \quad \psi_{1,k}(1, i_1)\beta_k^\top = -\beta_k^\top B(i_1), \quad (11b)$$

where $\psi_{0,k}(i_0, i_1)$ rotates column vectors of $\beta_k^\top$ as if they are circularly shifted.

Figure 2: Illustrations for (10) for $g = (0, 1)$ at left and $g = (1, 1)$ at right. Red, light yellow and green denotes negative, 0 and positive values, respectively.

## 2.5 HARMONIC FILTERS

Weiler et al. (2018) proposes the condition of a filter kernel $\kappa$ to be equivariant under the action $g \in G$.

**Lemma 1.** *The map $f \mapsto \kappa \cdot f$ is equivariant under $G$ if and only if for all $g \in G$,*

$$\kappa(gx) = \rho_{out}(g)\kappa(x)\rho_{in}(g)^{-1}. \tag{12}$$

Weiler & Cesa (2019) proves that such filters can be denoted by a series of harmonic bases $b(\phi)$, i.e.

$$\kappa(r, \phi) = \sum_{b \in \mathcal{K}} \omega_b(r)b(\phi), \tag{13}$$

where $\omega_b(r)$ is the per radial weights and $\mathcal{K}$ is a set of harmonic bases as dervied in the appendix of Weiler & Cesa (2019). For example, consider $\rho_{\text{in}} = \psi_{i,m}$ and $\rho_{\text{out}} = \psi_{j,n}$ in $\text{D}_N$,

$$\mathcal{K}_{\psi_{j,m} \leftarrow \psi_{i,n}} = \left\{ b_{\mu,\gamma,s}(\phi) = \psi(\mu\phi)\xi(s) \big| \mu = m - sn, s \in \{\pm1\} \right\}. \tag{14}$$

## 3 MAIN RESULTS

(12) and (13) provide a general approach to verify and construct steerable CNN with different representations. In this section, we relate these theories with filter transform and show how to use filter transform to construct steerable filters with input/output of different representations.

For readers who are not interested in group theory and mathematical derivation of the theory connection, we highlight the key equations to construct steerable filters in boxes. It should not be difficult to implement steerable filters directly from these equations using any modern deep learning framework. Fig. 3 shows illustration for these equations.

In our derivation, we mainly consider the angular coordinate of polar coordinate functions $\kappa(r, \phi)$ and write them $\kappa(\phi)$. We will also frequently make use of the following property:

$$\kappa(\phi - \theta_0) = \kappa(\phi + \theta_0), \quad \kappa(\phi - \theta_i) = \kappa(\phi + \theta_{N-i}). \tag{15}$$

### 3.1 FROM TRIVIAL REPRESENTATION TO REGULAR REPRESENTATION

**Rotation Group** $\text{C}_N$  Consider the the rotating filter $\mathsf{K}$ and its reflected version $\overline{\mathsf{K}}$ which are commonly used in previous works, e.g. TI-Pooling, ORN, RotEqNet and RotDCF:

$$\mathsf{K}(\phi) = \begin{bmatrix} \kappa^0 & \kappa^1 & \cdots & \kappa^{N-1} \end{bmatrix}^\top, \quad \kappa^n(\phi) = \kappa(\phi - \theta_n),$$
$$\overline{\mathsf{K}}(\phi) = \begin{bmatrix} \overline{\kappa}^0 & \overline{\kappa}^1 & \cdots & \overline{\kappa}^{N-1} \end{bmatrix}^\top, \quad \overline{\kappa}^n(\phi) = \kappa(\theta_n - \phi). \tag{16}$$

The output of convolution with the above kernels naturally permutes as the input rotates in $\text{C}_N$. This intuitively corresponds to property of a steerable filter transforming from trivial representation to regular representation. In this paper, we use $\mathsf{K}$ and $\overline{\mathsf{K}}$ as the basic filters to construct different types of steerable filters in $\text{C}_N$ and $\text{D}_N$. We verify the observation of the above steerability by substituting $\mathsf{K}$ into the lhs of Lemma 1 with $g = (0, 1)$ and write:

$$\mathsf{K}(\phi + \theta_1) = \begin{bmatrix} \kappa(\phi + \theta_1) & \kappa^0 & \cdots & \kappa^{N-2} \end{bmatrix}^\top = \begin{bmatrix} \kappa^{N-1} & \kappa^0 & \cdots & \kappa^{N-2} \end{bmatrix}^\top \tag{17a}$$

$$= \rho_{\text{reg}}^{\text{C}_N}(0, 1)\mathsf{K}\rho_{\text{tri}}(0, 1)^{-1}. \tag{17b}$$

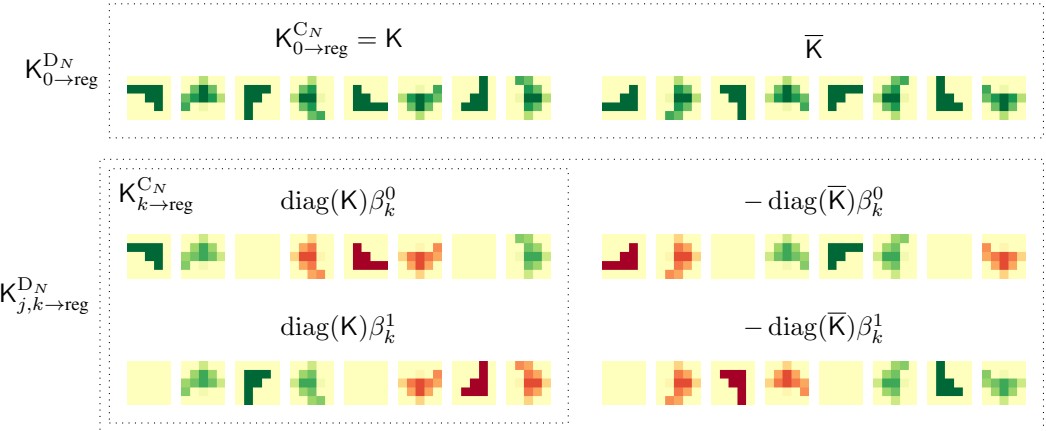

Figure 3: Visualization of FILTRA filter examples. Based on a same weight kernel $\mathsf{K}$, we generate filters $\mathsf{K}^{C_N}_{0\to\text{reg}}$, $\mathsf{K}^{D_N}_{0\to\text{reg}}$, $\mathsf{K}^{C_N}_{k\to\text{reg}}$ and $\mathsf{K}^{D_N}_{j,k\to\text{reg}}$. In this example we set $j=1, k=1, N=8$. The two-columns of matrix $\beta_k$ is splitted as $\beta_k^0$ and $\beta_k^1$ for visualization. Red, light yellow and green denotes negative, 0 and positive values, respectively. Please view this figure in color.

The above equation can be similarly verified for other $g = (0, i_1)$ and also on $\overline{\mathsf{K}}$. Thus WLOG we select the steerable filter which transforms trivial representation to regular representation on $C_N$ as

$$\boxed{\mathsf{K}^{C_N}_{0\to\text{reg}} = \mathsf{K}.} \tag{18}$$

**Dihedral Group** $D_N$   The steerable filter that transforms trivial representation to regular representation on $D_N$ can be constructed as

$$\boxed{\mathsf{K}^{D_N}_{0\to\text{reg}}(\phi) = \left[\mathsf{K}^\top \quad \overline{\mathsf{K}}^\top\right]^\top,} \tag{19}$$

which corresponds to enumerating each $D_N$ element and act on the kernel $\kappa$. For $g = (0, i_1)$, $\mathsf{K}^{D_N}_{0\to\text{reg}}$ can be verified to follow (12) in the same way as (17a), i.e. $\mathsf{K}^{D_N}_{0\to\text{reg}}(\phi+\theta) = \rho^{D_N}_{\text{reg}}(g)\mathsf{K}^{D_N}_{0\to\text{reg}}\rho_{\text{tri}}(g)^{-1}$.

For reflected action, when $g = (1, 1)$, we write:

$$\begin{aligned}
\mathsf{K}(-\phi + \theta_1) &= \left[\kappa(-\phi+\theta_1) \quad \kappa(-\phi-\theta_0) \quad \kappa(-\phi-\theta_1) \quad \cdots \quad \kappa(-\phi-\theta_{N-2})\right]^\top \\
&= \left[\overline{\kappa}^1 \quad \overline{\kappa}^0 \quad \overline{\kappa}^{N-1} \quad \cdots \quad \overline{\kappa}^2\right]^\top = B(1)\overline{\mathsf{K}}.
\end{aligned} \tag{20}$$

Similarly, we can show for $g = (1, i_1)$,

$$\overline{\mathsf{K}}(-\phi + \theta_{i_1}) = B(i_1)\mathsf{K}, \quad \mathsf{K}(-\phi + \theta_{i_1}) = B(i_1)\overline{\mathsf{K}}. \tag{21}$$

Thus we verify (12) for the reflected actions $g = (1, i_1)$ by summarizing the above as $\mathsf{K}^{D_N}_{0\to\text{reg}}(-\phi + \theta_{i_1}) = \rho^{D_N}_{\text{reg}}(g)\mathsf{K}^{D_N}_{0\to\text{reg}}\rho_{\text{tri}}(g)^{-1}$.

## 3.2   FROM IRREP TO REGULAR REPRESENTATION

**Rotation Group** $C_N$   Consider a $C_N$ irrep $\psi_{0,k}(g)$ with frequency $(0, k)$. We show that the following kernel

$$\boxed{\mathsf{K}^{C_N}_{k\to\text{reg}} = \text{diag}(\mathsf{K})\beta_k,} \tag{22}$$

transforms from $\psi_{0,k}(g)$ to regular representation for the action $g = (0, i_1)$. The derivation of correctness can be found in the appendix.

**Dihedral Group** $D_N$    Consider a $D_N$ irrep $\psi_{j,k}(i_0, i_1)$ with frequency $(j, k)$. We show that the following kernel:

$$\mathsf{K}^{D_N}_{j,k \to \text{reg}} = \left[ \mathsf{K}^{C_N \top}_{k \to \text{reg}} \quad (-1)^j \cdot \overline{\mathsf{K}}^{C_N \top}_{k \to \text{reg}} \right]^\top \tag{23}$$

transforms from $\psi_{j,k}(i_0, i_1)$ to regular representation for the action $g = (i_0, i_1) \in D_N$.

## 3.3 From Regular Representation to Regular Representation

Regular representation possesses a nice property that it can be averaged, pooled or activated channel-wise without violating steerability (Weiler & Cesa, 2019). Thus it is convenient to used regular representation for the intermediate features of a steerable CNN. We show in this subsection that the following kernels can be use to construct a steerable kernel whose input and output features are both in regular representation.

**Rotation Group** $C_N$

$$\mathsf{K}^{C_N}_{\text{reg} \to \text{reg}} = \left[ \mathsf{K}^{C_N}_{0 \to \text{reg}} \cdots \mathsf{K}^{C_N}_{\lfloor \frac{N}{2} \rfloor \to \text{reg}} \right] V^{-1}. \tag{24}$$

**Dihedral Group** $D_N$

$$\mathsf{K}^{D_N}_{\text{reg} \to \text{reg}} = \left[ \mathsf{K}^{D_N}_{0,0 \to \text{reg}} \cdots \mathsf{K}^{D_N}_{0,\lfloor \frac{N}{2} \rfloor \to \text{reg}} \quad \mathsf{K}^{D_N}_{1,0 \to \text{reg}} \cdots \mathsf{K}^{D_N}_{1,\lfloor \frac{N}{2} \rfloor \to \text{reg}} \right] W^{-1}. \tag{25}$$

The above two kernels can be verified to transform regular representation to regular representation in similar way and we show the derivation for the $C_N$ case (25) as an example in the appendix.

## 3.4 Reversed Transform of Representations

It is obvious to find that for (12), if $\rho_{\text{in}}, \rho_{\text{out}}$ are orthogonal matrices, i.e. $\rho_{\text{in}}^{-1} = \rho_{\text{in}}^\top, \rho_{\text{out}}^{-1} = \rho_{\text{out}}^\top$, the transpose of (12) naturally proves the equivariance of $\kappa^\top$ under a reversed representation transform direction, i.e. from $\rho_{\text{out}}$ to $\rho_{\text{in}}$. Thus we can easily obtain equivariance kernel from regular representation to trivial/irreducible representation by simply transposing (18), (19), (22) and (23).

## 3.5 Conventional Rotating Filters

We comprehensively study the approach to use filter rotation to form steerable convolutional kernels with regular representation features as input or output. Conventional filter rotation based networks exploit some basic forms introduced in this section. TI-Pooling (Laptev et al., 2016) exploits kernel $\mathsf{K}^{C_N}$ to transform trivial to regular representation, executes orientation pooling to convert regular to trivial representation and loses orientation information. RotDCF and ORN exploits a kernel of form

$$\mathsf{K}^{C_N}_{\text{ORN}} = \texttt{circulant}(\mathsf{K}). \tag{26}$$

It is easy to verify that $\mathsf{K}^{C_N}_{\text{ORN}}$ also follows Lemma 1 to be a steerable filter. However, compared to $\mathsf{K}^{C_N}_{\text{reg} \to \text{reg}}$, $\mathsf{K}^{C_N}_{\text{ORN}}$ consumes same filter storage but has less weight capacity ($N$ v.s. $N \lfloor \frac{N}{2} \rfloor$). RotEqNet constructs 2D vector field which could rotate as its input rotates but regards the 2D vector field as independent trivial representation in convolution. As shown in this paper, it preserves better steerability to regards the vector field as irrep representation with frequency 1.

## 3.6 Numerical Accuracy for Discrete Kernels

Note that when implementing discrete convolution, the equality of (17a) does not perfectly hold. For example, consider $\kappa^n(\phi) = \kappa(\phi - \theta_n)$, $\kappa^n(\theta_n) = \kappa(0)$ holds for a continuous $\kappa$. However, for discrete $\kappa$, $\kappa^n(\phi)$ is a rotated interpolation of $\kappa(\phi)$ and this equality does not precisely hold in general. There exist some exceptions where the equality can be achieved for discrete $\kappa$. One example is when $\kappa^n(\phi)$ is a $90°$ rotation of $\kappa$ and it can be precisely constructed from $\kappa$. Another example is when $\kappa^n$ is a $45°$ rotation interpolated by nearest pixel from a $\kappa$ of size $3 \times 3$.

| layer | k | s | output | $\delta t$ (FIL) | $\delta t$ (R2) |
|---|---|---|---|---|---|
| conv+relu | 5 | 1 | 128 (reg) | 0.12 | 0.17 |
| conv+relu | 5 | 1 | 192 (reg) | 0.13 | 0.13 |
| pool | 3 | 2 | 256 (reg) | - | - |
| conv+relu | 5 | 1 | 256 (reg) | 0.13 | 0.13 |
| conv+relu | 5 | 1 | 384 (reg) | 0.23 | 0.23 |
| pool | 3 | 2 | 384 (reg) | - | - |
| conv+relu | 5 | 1 | 512 (reg) | 0.32 | 0.48 |
| conv+relu | 5 | 1 | 768 (reg) | 0.62 | 0.91 |
| pool | 3 | 2 | 768 (reg) | - | - |

(a) The backbone network structure used in our experiments is composed by convolution, ReLU and pooling layers. The convolution layers are realized by FILTRA, R2Conv and conventional convolution respectively while the rest layers remain the same. Three realizations have the same number of output channels in each layer but organize the channels to be follow regular representation for FILTRA and R2Conv. k: kernel size. s: stride. $\delta t$: filter generation time in ms.

| layer | k | s | output |
|---|---|---|---|
| GroupPool | – | 1 | 24 (reg) |
| fc+relu | – | – | 16 (reg) |
| fc | – | – | 10 (tri) |

(b) The classification head network structure used in our experiments uses a Grouping Pooling (Weiler & Cesa, 2019) to generate transform invariant features.

| layer | k | s | output |
|---|---|---|---|
| PAMaxPool | – | – | 24 (reg) |
| conv+relu | 1 | 1 | 16 (reg) |
| conv | 1 | 1 | 2 (irrep) |

(c) The regression head network structure used in our experiments uses a PointwiseAdaptiveMaxPool (PAMaxPool) (Weiler & Cesa, 2019) to summarize feature in regular representation.

Table 1: Network structure in experiments

## 3.7 STEERABLE CNN WITH MULTIPLE LAYERS

A conventional CNN is usually composed convolution, pooling, nonlinearity and fully-connected layers. To achieve equivariance for the overall network, it is desired that all the component layers are steerable. As analyzed in the appendix of Weiler & Cesa (2019), channel-wise nonlinearity and channel-wise pooling preserves the steerability on feature maps with regular representation. fully-connected layers is a special case of convolution with $1 \times 1$ kernels and thus can be easily realized by steerable convolution.

## 4 EXPERIMENTS

The proposed equivariant convolution, refered as FILTRA, can be interpreted as an alternative formulation for the harmonic based (Weiler & Cesa, 2019) implementation of steerable convolution. In this section we show the pros and cons of each implementation by experiments. We make use of the framework E2CNN (Weiler & Cesa, 2019) for our experiments as it provides the general interface and operations for steerable CNN network. Experiments are executed on the MNIST, KM-NIST (Clanuwat et al., 2018), FashionMNIST (Xiao et al., 2017), EMNIST (Cohen et al., 2017) and CIFAR10 datasets.

We compare FILTRA against two convolution operations, i.e. the representative harmonic based convolution R2Conv (Weiler & Cesa, 2019) from E2CNN and the conventional vanilla convolution. All MNIST-like datasets are experimented on a same feature extraction backbone as described in Table 1a, with convolution operator realized by the three experimented approaches. CIFAR10 is experimented with WideResNet (Zagoruyko & Komodakis, 2016) in the setting similar to Weiler & Cesa (2019). We found that on CIFAR10, $C_4$ steerable network performs better than $C_8$ for both approaches. For all experiments, we randomly rotate or reflect according to the experiment settings. The settings and evaluation results are listed in Table 2. Different from Weiler et al. (2018), we force the three convolution kernels to output same number of channels. For example, compared to vanilla convolution, the number of free weights for a $C_8$ FILTRA is reduced to $1/8$ and for a $D_8$ is reduced to $1/16$. The filters for all the approaches will thus have exactly same shape at the deploy stage.

Experiments are executed on GTX 2070. The training procedure of FILTRA and R2Conv can both be implemented as a vanilla convolution plus a filter generation step. For $C_8$ case the runtime of both generator is similar and for $D_8$ case FILTRA is slightly faster. We show runtime of $D_8$ case in Table 1a at training stage. R2Conv additionally requires a initialization of about 2 min. Both of the approaches consume the same inference time as of vanilla convolution.

| Tasks | Classification (acc) | | | | | | | | | | Regression (angle err deg) | | | | | | | |
|---|---|---|---|---|---|---|---|---|---|---|---|---|---|---|---|---|---|---|
| | mnist | | kmnist | | fmnist | | emnist | | cifar10 | | mnist | | kmnist | | fmnist | | emnist | |
| Aug | S | O | S | O | S | O | S | O | wrn | wrn | S | O | S | O | S | O | S | O |
| Net eqiv | $C_8$ | $D_8$ | $C_8$ | $D_8$ | $C_8$ | $D_8$ | $C_8$ | $D_8$ | $C_4$ | $D_4$ | $C_8$ | $D_8$ | $C_8$ | $D_8$ | $C_8$ | $D_8$ | $C_8$ | $D_8$ |
| FILTRA | 98.9 | 98.1 | 97.1 | 97.0 | 90.5 | 90.8 | 77.1 | 80.5 | 93.4 | 92.8 | 3.3 | 5.4 | 3.2 | 3.6 | 2.6 | 2.8 | 29.8 | 24.9 |
| R2Conv | 98.8 | 98.1 | 97.3 | 96.8 | 90.5 | 90.8 | 76.7 | 80.1 | 93.6 | 92.7 | 4.8 | 8.9 | 3.4 | 4.5 | 2.9 | 3.7 | 34.5 | 29.2 |
| Conv | 98.5 | 98.0 | 96.4 | 95.2 | 89.3 | 88.3 | 72.6 | 80.1 | 93.2 | - | 6.6 | 10.6 | 4.8 | 6.4 | 3.1 | 3.6 | 37.4 | 25.5 |

Table 2: Performance on MNIST and CIFAR10. S: randomly augmented over $SO(2)$. O: randomly augmented over $O(2)$. wrn: WideResNet. Zagoruyko & Komodakis (2016).

### 4.1 CLASSIFICATION TASK

The most typical experiment used in previous works on conventional steerable CNN is the classification task. We follow this convention and compare the classification performance of the experimented three approaches in Table 2. FILTRA show comparable performance to R2Conv and slightly improves accuracy for OCR-like (*MNIST) tasks where high frequency texture is limited. On CIFAR10, the performance of FILTRA is minorly disadvantageous. The explanation comes in the interpolation artifacts mentioned in Subsect. 3.6. As the interpolation of high frequency components deviates more, this harms the performance on CIFAR10 with high frequency texture.

### 4.2 REGRESSION TASK

Besides the typical classification task, we find that the property of steerability is naturally advantageous for many regression tasks whose input might rotate or reflect. In this paper, we evaluate the regression performance with an example task to predict the character direction. Similar tasks are commonly used in OCR techniques. When the character rotates, the predicted direction should rotate with the same rotating frequency. This means the predicted 2D direction vector is following a irrep $\psi_{0,1}$ for $C_N$. We reuse the backbone in Table 1a to extract features and use a regression head in Table 1c to predict a unit 2D vector denoting the direction. The network is trained with MSE loss. Note that the images should be masked by a disk to avoid the network to overfit the direction from rotated black boundary. Different approaches are evaluated by the mean included angle between the predicted and groundtruth directions as shown in Table 2. FILTRA with $C_8$ steerability performs best when trained on data augmented over $SO(2)$. We owe this to the fact that FILTRA weight is naturally organized by the discrete grid layout. Each element of discrete weight matrix contribute to one more DoF of the filters. In contrast, R2Conv uses filters parameterized with a polar coordinate. The DoF of the filters is slightly reduced due to the discretization.

## 5 CONCLUSIONS

In this paper, we establish the connection between the recent steerable CNN structure based on group representation theory and the conventional transformed filters. To this end, we propose an approach to construct steerable convolution filters, which transform between features in trival, irreducible and regular representations. We verify the feasibility of FILTRA for the classification and regression tasks on several datasets.

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
