# OpenReview forum: "FILTRA: Rethinking Steerable CNN by Filter Transform"
_ICLR.cc/2021/Conference — Reject_

### Official Review · AnonReviewer4 · 2020-10-22

**Rating:** 6
**Confidence:** 4

**Review:**

Equivariant Steerable CNNs for 2D/3D rotation+reflection+translation groups have generally been implemented as a filter transform/expansion step followed by a standard convolution. The filter expansion step involves taking a linear combination of steerable basis filters. These basis filters are pre-computed before network training by solving a linear system or by sampling the continuous analytical solution (this can take a few minutes). Depending on the chosen group representation wrt which the network layer is equivariant, a different filter basis will emerge, but in general one can see that the basis filters come out as rotated and flipped copies of some basis filters, with the occasional sign flip (this has been stated in some earlier works). The precise way in which a basis filter is to be rotated and flipped to obtain the steerable filter basis for the 2D case had not been worked out before, to my knowledge, and this is one of the contributions of this paper. The analysis is done for each (input, output) representation type chosen from {trivial, irreducible, regular} representations. Having worked this out, the paper proposes to use this as a way of implementing the filter expansion step, starting from a basis filter and rotating/flipping it to obtain an expanded filter bank.

The proposed method (FILTRA) does not require a precomputation step if I understand correctly, which is a significant practical advantage. Experiments further show that the method is similar or faster at filter expansion. Finally, the method is validated by training networks on benchmark tasks and shown to perform similarly to or better than the steerable CNN implementation of Weiler & Cesa, which is the best existing implementation.

The paper briefly mentions that filters cannot be rotated exactly on a discrete grid, but I didn't figure out how the authors propose to deal with this issue. How exactly are the filters rotated?

I think the method proposed in the paper is useful, as it is both faster and better than existing steerable CNN implementations. The paper itself is fairly well written and technically correct as far as I can tell, but may be challenging to read for those who are not yet knowledgeable about steerable CNNs. Those readers however are unlikely to be interested in learning about the implementation details of steerable CNNs anyway, so perhaps this is fine. The reason I am not giving a higher rating is that I think that although this is a useful contribution to the literature on steerable CNNs, which are being used in an increasing number of applications, the paper does not represent a major breakthrough and although the calculations are non-trivial, does not contain highly unexpected or deep theoretical results.

Typos:
Cadestrian -> Cartesian
irreduciable -> irreducible
equity -> equality

Edit:
Having read the reviews, rebuttal and updated paper, I have decided to maintain my score of 6.

---

> ### Author Response · Authors · 2020-11-24
> **Authors' Response to AnonReviewer4**
>
> **Q4.1: The paper briefly mentions that filters cannot be rotated exactly on a discrete grid, but I didn't figure out how the authors propose to deal with this issue. How exactly are the filters rotated?**
>
> A4.1: Section 3.6 mentions an error source for filter transform. However, in our implementation this is not specifically handled since the overall performance does not drop too much.
>
> **Q4.2: The reason I am not giving a higher rating is that I think that although this is a useful contribution to the literature on steerable CNNs, which are being used
> in an increasing number of applications, the paper does not represent a major breakthrough and although the calculations are non-trivial, does not contain highly unexpected or deep theoretical
> results.**
>
> A4.2: We present a practical and elegant research on steerable filter that is both novel and non-trivial to derive. Compared to the previous harmonic-based steerable filters, the boxed formulations proposed in our paper are very simple and elegant. Nonetheless, the construction of the formulations, especially between the irrep and regular representation, are
> non-trivial and requires high mathematical skills. The simplicity and practicality of our approach eases the effort needed for the CV community to make use of steerable CNN in practical applications. We
> believe this will significantly contribute to the impact of steerable CNNs.

---

### Official Review · AnonReviewer1 · 2020-10-26
**Initial review for FILTRA**

**Rating:** 5
**Confidence:** 5

**Review:**

***Summary***

The paper outlines the construction of filters in a group equivariant convolutional network equivariant to the groups CN (cyclic group of order N (N rotations)) and DN (dihedral group of order N (N rotations and flipping)). The authors achieve this by taking linear combinations of (harmonic) basis filters, as was shown in Weiler et al. (2018). They then proceed to show explicitly how one can build filters between particular representations of each group, namely, the trivial, irreducible, and regular representations. For experiments they demonstrate that the filters perform en par with those of Weiler and Cesa (2019), who had performed a near-exhaustive comparison among representations of SE(2, R).



***Pros***

Technically I believe the work to be sound. I did not see any mistakes jump out at me. I also think that it is laudable that the authors included a high amount of detail in their exposition, which lays bare the exact mechanisms by which one would effectively go about building an equivariant layer.

Experimentally it appears that the authors chose a sensible baseline and that they compared on dataset, which are relevant to the topic and commonly used in the equivariance literature. The authors also provide timing information on how fast each filter can be generated, which is something I have not seen before in the equivariance literature and which I appreciate seeing.




***Cons and constructive feedback***

Perhaps my largest criticism is that it is not obvious exactly what the contribution of the paper is meant to be. Perhaps this originates from my not knowing what the authors mean by “filter transform” and “steerable CNN”. My understanding is that the authors believe it is currently unknown how to construct a steerable CNN for the cyclic and dihedral groups. They provide explicit constructions in section 3, but to my knowledge these are already provided in the long appendix of Weiler and Cesa (2019), furthermore they can be found in Cohen and Welling (2016), Cohen and Welling (2017) and Bekkers et al. (2018) and most notably Weiler et al. (2018), who were to first to show how to construct equivariant filter layers some linear combinations of harmonic basis filters. As a result I am unsure of how to gauge the novelty of this contribution.

In the experiments of Table 2 (classification and regression) the authors compare their framework with an R2conv (representative harmonic based convolution) and a regular translationally equivariant network. I would like to know what exactly is a representative harmonic based convolution? Is that meant to refer to the work of Weiler and Cesa (2019)?

In the results tables there are no error bars. If possible I would have liked to have seen them. Since they are not there is it very hard to judge the efficacy of the results, which differ from the baselines by very small amounts.

Given the proximity of the work to Weiler and Cesa (2019), I would like to know exactly what differentiates the two works.

Please include exactly what the functions roll, flipped, and circulant actually do mathematically. I found these difficult to parse.

The mathematical level of the paper is pretty heavy for the uninitiated. It may be advisable for the authors to include a glossary of terms, if not short descriptions, in an appendix. If this is too much, at least please point to other papers, which have easily readable background sections for those not already well-read in group theory.

The authors may wish to have the submission proofread for spelling and grammar.

***Post rebuttal review***

Having read through the rebuttal, the updated submission, and the reviews of the other reviewers I have upgraded my review from a reject to marginally below acceptance. This is for two main reasons. 1) the authors have vastly improved the presentation of the submission, which now looks a lot easier to read, 2) the authors have clarified for me, at least, what the main contribution of the work is.

That said I am not entirely sure what this contribution adds to the equivariance community, hence why my recommendation still leans towards reject. As far as I am aware, solving the equivariance equations is not the large bottleneck to progress in our community. They are linear equations, and there is work back into the 80's solving them (check out people like Pietro Perona, Patrick Teo). I think more importantly we need to focus on pushing the boundaries in areas such as extension to non-Euclidean manifolds, convolution over non-compact groups, learning symmetries, etc. While this work is clearly mathematically sound and the authors have demonstrated deep knowledge of the area, it feels a little like retracing prior works. That said, if the other reviewers disagree then I don't mind this paper being accepted. Perhaps since I have worked in this area, what appears as obvious to me is not generally acknowledged and this paper may serve as a useful clarification for those wishing to dive into the literature.

---

> ### Author Response · Authors · 2020-11-23
> **Authors' Response to AnonReviewer1**
>
> **Q1.1: Perhaps this originates from my not knowing what the authors mean by “filter transform” and “steerable CNN”. My understanding is that the authors believe it is currently unknown how to construct a steerable CNN for the cyclic and dihedral groups. They provide explicit constructions in section 3, but to my knowledge these are already provided in the long appendix of Weiler and Cesa (2019), furthermore they can be found in Cohen and Welling (2016), Cohen and Welling (2017) and Bekkers et al. (2018) and most notably Weiler et al. (2018), who were to first to show how to construct equivariant filter layers some linear combinations of harmonic basis filters. As a result I am unsure of how to gauge the novelty of this contribution.**
>
> A1.1: Reviewer 1 interprets the term "steerable CNN" in the same way as it is in used in our paper. As mentioned in the second paragraph of Section 1, there are roughly two different categories of approaches to construct steerable filters. The first category solves equivariant filter (harmonic) basis and construct steerable filters by linear combination. Cohen and Welling (2016), Cohen and Welling (2017) and Weiler and Cesa (2019) are all representative works in this category, and we are fully aware of these works as mentioned in our paper. The term "filter transform" refers to the other category which constructs steerable filters from an normal filter by duplicating its rotated copies. Bekkers et al. (2018) also falls in this category.
>
> Both categories have been long studied in the past years. The filter transform approach is more widely known in the CV community due to the simplicity on both theory and implementation. However, previous filter transform approaches fail to produce completed steerability under different group representations, which is notably solved in Weiler and Cesa (2019) using harmonics filters. The aim of our paper is to complete the puzzle for filter transform approaches to also achieve a completed steerability for different group representations i.e. trivial, irreducible and regular representations. This provides an alternative approach to construct steerable filters which is mathematically equivalent, precomputation-free, faster in runtime and also straightforward to implement (with core code in 100L pytorch) without requiring knowledge of harmonic mathematics. We believe it helps push the practical application of steerable CNNs in CV to bridge the two categories of approaches.
>
> **Q1.2: In the experiments of Table 2 (classification and regression) the authors compare their framework with an R2conv (representative harmonic based convolution) and a regular translationally equivariant network. I would like to know what exactly is a representative harmonic based convolution? Is that meant to refer to the work of Weiler and Cesa (2019)?**
>
> A1.2: Yes. In our paper, E2CNN (package) and R2Conv (operator) both refer to the work of Weiler and Cesa (2019).
>
> **Q1.3: Given the proximity of the work to Weiler and Cesa (2019), I would like to know exactly what differentiates the two works.**
>
> A1.3: As mentioned in A1.1, the propoased filter transform approach is an alternative approach to construct steerable filters without using harmonic bases. It is mathematically equivalent to harmonic based approach e.g. Weiler and Cesa (2019), but are faster in runtime, precomputation-free and also straightforward to implement.
>
> **Q1.4: Please include exactly what the functions roll, flipped, and circulant actually do mathematically. I found these difficult to parse.**
>
> A1.4: We have added more explanation and reference for these operators in our revised paper. As mentioned in Section 2, these functions are borrowed from NumPy or SciPy to manipulate matrix.
>
> **Q1.5: The mathematical level of the paper is pretty heavy for the uninitiated. It may be advisable for the authors to include a glossary of terms, if not short descriptions, in an appendix. If this is too much, at least please point to other papers, which have easily readable background sections for those not already well-read in group theory.**
>
> A1.5: Our paper heavily re-uses the notations from Weiler & Cesa 2019 to help readers keep track with all the concepts. We have added more recapitulation of these concepts in the revision to ease readers who are not familiar with Steerable CNN.
>
> **Q1.6: The authors may wish to have the submission proofread for spelling and grammar.**
>
> A1.6: Thanks for the suggestion. We will proofread the final version.

---

### Official Review · AnonReviewer2 · 2020-10-28
**This paper provide the theoretical connection between steerable convolution and filter transform method.**

**Rating:** 6
**Confidence:** 3

**Review:**

This paper studies the connection between steerable CNN and filter transformation. The authors show theoretically that filter transformation can be used to realize steerable filters over different group representations. The authors also empirically show that the filter transformation based steerable CNN performs on par with the implementation based on harmonic bases.

Overall, I think this paper is hard to follow. The presentation and notation depends on that of prior works and is hard to understand without being familiar with Steerable CNN. Compared with prior works, this paper focuses more on the mathematical derivation but falls short of explaining their implication. The result is therefore hard to understand for those that are not familiar with the theoretical basis. Also, the authors do not clearly show why the contribution of this work is significant. The theoretical connection doesn’t seem to provide a significant advantage nor solve the problems in existing methods. Similarly, the experiment doesn’t provide much information and only shows that the proposed implementation works on simple datasets. The authors should try to highlight how this work may benefit other research or applications and verify that the proposed method works on more complicated and realistic data.

Another aspect that can be improved is to discuss how the theory generalizes when multiple convolution layers are applied. In general, the theoretical properties of steerable convolution does not automatically generalize to multiple convolution layers. This is particularly important in real world data, because the transformation may happen at different scales and may need to be accounted for at different layers in the network. Therefore, the results derived from a single convolution operation may not be sufficient.

---

> ### Author Response · Authors · 2020-11-23
> **Authors' Response to AnonReviewer2**
>
> **Q2.1: The presentation and notation depends on that of prior works and is hard to understand without being familiar with Steerable CNN. Compared with prior works, this paper focuses more on the mathematical derivation but falls short of explaining their implication. The result is therefore hard to understand for those that are not familiar with the theoretical basis.**
>
> A2.1: It is indeed difficult to explain all the mathematical concepts related group equivariance in a paper with limited length. Therefore, our paper heavily re-uses the notations from Weiler & Cesa 2019 to help readers keep track with all the concepts. We have added more recapitulation of these concepts in revision to ease readers who are not familiar with steerable CNN. The main results (Section 3) in our paper are organized by two components: an equation to construct steerable kernels (wireframed boxes) and a derivation that the kernels follows Lemma 1. The implication of most mathematical derivation is just the proof of correctness. For readers who are not interested in the math part, boxed equations are the take-away tips which can be directly used to construct kernels. An intuitive illustration of the constructed kernels is also shown in Figure 3.
>
> **Q2.2: The authors do not clearly show why the contribution of this work is significant. The theoretical connection doesn’t seem to provide a significant advantage nor solve the problems in existing methods.**
>
> A2.2: As mentioned in our paper, steerable CNN is mainly reconstructed in two approaches in previous methods, i.e. by transforming/rotating kernels and by solving steerable bases. The former are easy to understand, widely known in CV community but only produces limited steerability in previous works. The latter produces complete steerability in function space but requires more mathematical knowledge to deploy in applications. Our paper not only aims to bridge the two approaches, but also provides an easier approach for CV community to implement steerable CNN without worrying about the harmonic function theory behind. Our approach is mathematically equivalent, faster in runtime and also straightforward to implement (with core code in in 100L pytorch).
>
> **Q2.3: The experiment doesn’t provide much information and only shows that the proposed implementation works on simple datasets. The authors should try to highlight how this work may benefit other research or applications and verify that the proposed method works on more complicated and realistic data.**
>
> A2.3: As mentioned in the introduction, the application of steerable CNN ranges over OCR, aerial image processing, 3D point cloud processing, medical image processing, texture analysis, etc. For conventional classification and regression tasks, MNIST with rotation reflects the application in practical OCR on rotated documents and is the mostly widely used dataset in previous works on steerable CNN. We further extend MNIST to X-MNIST dataset on different character set. Conventional images such as CIFAR is also experimented in our paper, but we think this is not a representative scenario where steerable CNN is essential. This is because the images are mostly considered up-right in the practical application of generic image recognition. In addition, we would like to mention that it is interesting to consider steerable CNN in higher-level applications like objection detection or segmentation. This is a novel and attractive aspect, but beyond the scope of our paper.
>
> **Q2.4: Another aspect that can be improved is to discuss how the theory generalizes when multiple convolution layers are applied.**
>
> A2.4: Yes. As also mentioned or implied in previous works, e.g. Weiler & Cesa 2019, if all layers in a CNN is equivariant to group $g$, the overall network is equivariant to $g$. We have added an extra sub-section to clarify this question in the revision. In our experiment, we build equivariant multi-layer CNNs using equivariant activation and pooling layers, and evaluate its performance. As mentioned in A3.5, the regression task also specifically illustrates the equivariance of the network under transformed inputs.

---

### Official Review · AnonReviewer3 · 2020-10-29
**An interesting paper that builds the connection between steerable CNN based on group representation theory and filter transformation**

**Rating:** 6
**Confidence:** 4

**Review:**

This paper builds the connection between various steerable CNN structures based on group representation theory and filter transformations. Using the discrete rotation and reflection group as an example, the paper establishes ways to construct steerable convolutional filters that transform features in trivial, regular, and irreducible representations. In particular, recent works on steerable CNNs such as ORN, RotDCF, TI-pooling, and RotEqNet can all be explained under such framework.

The paper is generally well written and well organized. However, I reckon the material would be quite dense for readers interested in equivariant CNNs but not well-versed in group representation theory, so a bit more explanation either in text or in the appendix would be helpful. The main theoretical results in the paper on the considered discrete group transformation are interesting, and can potentially lead to other development in the community. The numerical experiments seem to be limited (though this phenomenon seems to be the issue for most of the papers in this area...)

** Pros
1. Well written and technically correct.
2. Interesting theoretical results that might inspire other works in the field.

** Cons (to be explained in more details in questions)
1. Some notation and abbreviation are used without proper explanation, which might be confusing.
2. Limited experimental results

** Questions and comment
1. Some of the notations and abbreviations are used without proper explanation. For example, "K" in equation (12), OCR on page 1, and missing reference on R2Conv. I would encourage the authors to make a greater effort to clarify their ideas and results.

2. The theory seems to be built only on discrete group. Does it generalize to continuous group transformation (such as SO(2) before discretization)?

3. Also, does the theory generalize to non-compact groups such as  scaling and shearing?

4. The experiments seem to be limited, even though this seems to be a common issue in papers in this field. However, I do recommend the authors to present the "equivariant loss" when using their proposed FILTRA, considering that discretization and interpolation might cause a problem in their setting unlike other means of steerable CNNs such as RotDCF and Harmonic Net

---

> ### Author Response · Authors · 2020-11-23
> **Authors' Response to AnonReviewer3**
>
> **Q3.1: A bit more explanation either in text or in the appendix would be helpful.**
>
> A3.1: Yes. We have added more explanation in our revised paper.
>
> **Q3.2: Some of the notations and abbreviations are used without proper explanation. For example, "K" in equation (12), OCR on page 1, and missing reference on R2Conv. I would encourage the authors to make a greater effort to clarify their ideas and results.**
>
> A3.3: We have added more clarification to resolve the confusions in the revised paper. "K" in equation (14) comes as a supplement for equation (13). Both (13) and (14) is from Weiler & Cesa 2019. E2CNN and R2Conv refer to the same work of Weiler & Cesa 2019. E2CNN is the name of the whole package while R2Conv is the equivariant convolution operator.
>
> **Q3.3: The theory seems to be built only on discrete group. Does it generalize to continuous group transformation (such as SO(2) before discretization)?**
>
> A3.4: This is a very interesting question. Continuous group transformation requires separate mathematical analysis that are beyond the scope of our paper. We roughly answer this problem from two aspects: 1) Consider a regular representation on $C\_N$. The irreps in equation (9) has frequency of at most $N$. Thus if a signal rotates with angle less that $2 \pi / N$, these irreps might not be able to fully reconstruct the difference before/after the rotation. 2) Consider equation (17b) as an example, a fractional rotation between $i$ and $i + 1$ in $C\_N$ will produces from (9) a fractional matrix $\rho\_\text{reg}$ smoothing $\rho\_\text{reg}(i)$ and $\rho\_\text{reg}(i + 1)$. This matrix tries to interpolate $K(\phi + \theta)$ from $K$ with Fourier bases. However, this interpolation is only approximate due to the lack of high frequency bases.
>
> **Q3.4: Does the theory generalize to non-compact groups such as scaling and shearing?**
>
> A3.5: We notice some existing papers, e.g. Worrall & Welling 2019 are working on this problem. It should be possible if the discretization of scaling and shearing is restricted in a finite range. It is interesting to unify more transform groups in similar techniques.
>
> **Q3.5: The experiments seem to be limited, even though this seems to be a common issue in papers in this field. However, I do recommend the authors to present the "equivariant loss" when using their proposed FILTRA, considering that discretization and interpolation might cause a problem in their setting unlike other means of steerable CNNs such as RotDCF and Harmonic Net.**
>
> A3.5: The "equivariant loss" is also a metric we used in the experiments to evaluate performance. We use the regression task with mean angle error as the measurement of the equivariance. The input is rotated in by angles from SO(2) instead of $C\_N$. We think the angle loss reflects the direction consistency information between the input and output.

---

### Author Response · Authors · 2020-11-23
**Updates from the Authors**

We sincerely thank all reviewers for the constructive suggestion. A new version of our paper has been uploaded, with mainly the following revision:

* More explanation of mathematical basics on group representation and steerable filter have been added, including concepts, figures and references.
* We move some mathematical derivation to an extra appendix as uploaded in the supplementary materials.
* Other revision mentioned in the following response.

In the following, we respond to the questions reported by the reviewers. **All equations and figures mentioned in the response are refered by their numbers in our revised paper.**

---

### Decision · Program_Chairs · 2021-01-07
**Final Decision**

**Decision:**

Reject

**Comment:**

This paper introduces an approach based on filter transform for designing networks equivariant to different transformation groups.
Especially, the authors rely on the haramonic analysis view of steerable CNNs given in Weiler & Cesa (2019) to design an equivariant filter bank by computing simple transforms over base filters.

The reviewers finds the paper technically solid but difficult to read and with a limited contribution.
The AC carefully reads the paper and discussions. Although the connection between steerable CNNs and filter transform are interesting, the AC considers that the main contributions of the paper should be consolidated, especially the positioning with respect to Weiler & Cesa (2019). \
Therefore, the AC recommends rejection.